# Knowledge, attitudes, and practices (KAP) towards COVID-19 pandemic among pregnant women in a tertiary hospital in Karachi, Pakistan

Sumaira Naz, Syeda Dur e Shawar, Shamila Saleem, Ayesha Malik*, Amir Raza

Department of Obstetrics & Gynecology, Aga Khan University Hospital, Karachi, Pakistan

* ayesha.malik@aku.edu

## Abstract

### Background

The aim of this study was to evaluate the knowledge, attitude, and practices (KAP) of the pregnant population during the COVID-19 pandemic in a tertiary care hospital.

### Methods

This cross-sectional study was conducted at Aga Khan University Hospital, Karachi, Pakistan. KAP towards COVID-19 was assessed using 21-item questionnaires. A score for each category was calculated and points were summed. The outcome variables of KAP were compared with demographic characteristics. Data were analyzed by using SPSS 19.

### Results

A total of 377 patients participated in the study. The majority of the patients were multiparous (36.8%) in the age group of 30-40years (42.4%). More than 90% of patients were aware of COVID-19 symptoms and mode of transmission. They were aware of no cure for disease and optimum social distance. Although < 50% of patients truly answered the questions regarding the impact of COVID-19 on the risk of congenital malformation, vertical transmission, and the effect of infection on the mode of delivery. Regarding attitude and practices,> 90% of patients were anxious about fetal and personal safety, they are using a facemask, sanitizing their hands regularly, and avoiding social gatherings. Univariate and multivariable linear regression analysis showed statistically significant results among demographic variables (age, parity, family members, occupational status, and source of information).

### Conclusion

Pregnant patients demonstrated inadequate knowledge regarding the impact of COVID-19 on pregnancy. However positive attitude and practices on preventive measures were good.

**Data Availability Statement:** All relevant data are within the paper and its Supporting Information file.

**Funding:** The author(s) received no specific funding for this work.

**Competing interests:** The authors have declared that no competing interests exist.

This highlights the need for health education for pregnant women for COVID-19 to improve knowledge on a constant basis.

## Introduction

Coronavirus disease 2019 (COVID-2019), is an ongoing pandemic caused by a highly infectious novel virus COVID-19 belongs ta o large family of coronaviruses [1]. On January 31st, 2020, this outbreak has been declared a serious global health emergency by the World health organization (WHO) [2].

The course of illness ranging from mild flu-like illness to severe diseases like pneumonia, acute respiratory distress syndrome (ARDS), aseptic shock, and multi-organ failure [3]. Presently there is no specific treatment available for this disease [4]. Despite the implementation of vaccination, the third wave of this pandemic has gripped the whole globe and resulted in thousands of new cases and death all over the world [5].

Pregnancy is an immune-compromised state, which predisposes pregnant women to infection. Because of this sudden outbreak of COVID-19, lots of stress and anxiety have been created among pregnant women in different parts of the world [6]. The main reason for this anxiety and fear is the lack of provision of accurate information regarding pregnancy with COVID-19 infection and its complications [7]. In addition, there are concerns regarding the impact of COVID-19 infection on pregnancy, fetal, and neonatal outcomes all over the world [8].

Therefore, it is important to identify their worries and help them to reduce fear by providing correct and evidence-based information regarding COVID-19 disease and its sound effects on pregnancy [9].

Literature suggests that pregnant women with hypertensive, cardiovascular, and respiratory system diseases are at higher risk of maternal mortality in COVID-19 [10]. Although national data on COVID 19 related to pregnancy have been lacking. Hence, there is an urgent need for the evaluation of pregnant women's awareness regarding COVID-19 in developing countries like Pakistan to facilitate appropriate antenatal, intrapartum, and postpartum care. This Study explored the Knowledge, attitude, and practices (KAP) among pregnant patients attending during this COVID-19 outbreak.

## Materials and method

This cross-sectional study was conducted with a non-probability convenience sampling method over a 3-month period from February 15, 2022, to April 15, 2022, among pregnant women in obstetrics and gynecology department of a tertiary care hospital, Aga khan university hospital (AKUH) Karachi during an outbreak of covid-19. Data was collected after approval of the ethical review committee with the reference number; 2020-4976-12883 at Aga khan university hospital Karachi, Pakistan.

The sample size calculated by the WHO calculator is 360 assuming a response rate of 50%, confidence interval (CI) of 95%, Z as 1.96, and margin of error d as 5% by assuming 5000 deliveries occurred per annum in obstetrics and gynecology unit as per unit annual statistics. Hence, the sample size was n = (Z) $^2$ P (1-P) N /d$^2$ (N-1) + (Z) $^2$ P (1-P), by considering the incomplete responses, we included the target sample of 380 [11].

All pregnant patients attending clinics, wards, and labor rooms in a latent phase were invited to participate in the study, who agreed to participate, and informed consent in writing

has been taken. After the written consent study questionnaire was used to assess the Knowledge regarding COVID-19, also their attitude and Practices toward this outbreak. Patients not giving consent, who was in active labor, with acute emergencies were excluded.

All patients coming to the clinic had their vitals checked in the assessment Room. They were identified and approached in the assessment room and invited to participate in the study. Informed consent was taken in a history-taking room which was a separate room in the clinic.

Patients admitted to wards and labor room (latent phase), after taking Informed consent, are guided to a separate room (A2 teaching room located in between labor room and wards) to complete the questionnaire by themselves, whereas patients in the labor room (in latent phase) were approached in their assigned room (non-sharing) for consent. These rooms were used to ensure privacy and social distancing. One of the team members was available to facilitate the patients in case of queries related to the questionnaire. The time duration for filling one form was10–15 minutes.

Data was collected on the self-administered questionnaire, developed after a literature review and from WHO recommendation [12]. The questionnaire was prepared in English and then translated into the Urdu language with the assistance of language experts. The content of the questionnaire was grouped into various themes, including demographic characteristics, obstetric variables, knowledge, attitude, and practice of COVID-19-related questions. The questionnaire consisted of two parts: First part included demographic variables; age, parity, family members, education, source of information, employment, and area of residence. The age variable was categorized as: <30, 30–40, >40 years, and parity as a primigravida, para 1+0, para 2 to 4, and para 5+0 or more, whereas the number of households was taken as less than 4, 4–7, more than 7 members in family and education was categorized into no formal education, basic education (Matriculation), college and university level Employment status was considered an as a housewife and working. The area of residence was taken as rural and urban whereas the sources of information included health workers, friends and relatives, television, and the internet.

The second part included a total 21-item scale. Of the 21, twelve questions were regarding knowledge (clinical presentations, transmission routes, prevention of COVID-19) with additional 3 questions for attitude and 6 for practices against this outbreak.

Each question of the knowledge section had three options (Yes/No/ don't know). The correct answer was given 1 score and the incorrect answer was given a zero score. Overall knowledge scores ranged from zero to twelve. Individual scoring of 10 and above was categorized as excellent whereas scores below 10 were considered as poor knowledge.

The attitude section consisted of three questions, and the response of each item was collected on a 3-point Likert scale as follows 0 ("Disagree"), 1 ("Neutral"), and 2 ("Agree"). The total score ranged from 0 to 6 with an overall greater score indicating more positive attitudes toward the COVID-19. A cut-off level of 5–6 was set for more positive attitudes towards the prevention of COVID infection [13].

Similarly, the practice section included 6 items to assess practice measures related to the COVID-19, and each of the six questions was answered as Yes, No, and Sometimes. Practice items' total scores ranged from 0-to 6, with an overall greater score indicating more frequent practice towards the COVID-19. A cut-off level of ≥5 is set for more frequent practices [14].

The reliability of the questionnaire was assessed by using Cronbach's alpha and the Cronbach's alpha coefficient of the knowledge, attitude, and practice were 0.93, 0.98, and 0.85, respectively, and overall Cronbach's alpha of KAP questions was 0.93, which indicates acceptable internal consistency. No data from the pilot study were included in the final analyses. A pretest was conducted for understandability, language clarity, and relevance of the questionnaire among pregnant women by taking 10% of the sample size.

Data was entered and analyzed using SPSS version 19. Descriptive analysis was done by treating age, parity, and family members as continuous variables, whereas education was analyzed as ordinal data. Employment status, source of information, and residence were categorized as nominal data. Frequencies and percentages were calculated for categorical variables. Mean and standard deviation was calculated for continuous variables.

Scoring categories of knowledge, attitude, and Practices were compared with different categories of age, parity, number of family members, employment status, source of information, educational status, and area of residence. Chi-square test or fisher's exact test was used for Univariate analysis.

Multivariable linear regression analysis using all the demographic variables as independent variables and knowledge, attitude and practice score as the outcome variable were conducted to identify factors associated with knowledge. P≤0.05 was considered as significant.

## Results

A total of 380 women participated in the study however, we excluded three participants due to missing data. A total of 377 pregnant women have considered for the final analysis and the response rate was 99%. Of the total, 55.2% of women were below 30 years of age and 42.7% of pregnant women were of 31 to 40 years. More than 80% of participants belong to urban areas. More than 90% of the women had an education at the university or college level. Most of the women were housewives (76.4%). Out of 377 women, around 38.5% were grand multiparous and multiparous, while 23.3% of women were nulliparous. The main sources of information for these women were the internet (37.75%) and television (26%), while 20% of participants' source of information was health care workers (Table 1).

Less than 50% of the women responded that COVID19 infection developed the severe disease in pregnancy, it causes poor pregnancy outcome if occurs in the third trimester, affects the mode of delivery, it's increased the abnormality of the fetus, and increases the risk of vertical transmission. Fifty-five percent of women replied that breastfeeding is safe for mild COVID 19 infections. More than 90% of the participants were aware of the symptoms, spread of infection, an optimum social distancing between individuals, and lack of optimal treatment of COVID19. There were 85% of women responded that health care workers should wear an N-95 mask all the time while 75% of women were that patients without symptoms can transmit the infection to others. Overall responses of knowledge regarding the COVID 19 are illustrated in (Fig 1).

The average knowledge score was 6.63 ± 2.48 (Range: 0–12) (**Table 2**). There were only 13 (3.4%) women who had excellent knowledge (Score: 11–12), 78 (20.7%) women had good knowledge (Score: 9–10) and most of the study participants 286(75.9%) had poor knowledge (Score Range: 0–8) (**Fig 2**).

Nearly 88% of women had a positive attitude and the mean attitude score was 5.36±1.81 (Range: 0–6) as presented in (Table 2) & (Fig 2). Practices responses of women to limit the spread of COVID-19 are shown in (Table 3). Use of, a mask, sanitizer, and soap for hand washing, avoiding social gatherings, and maintaining a social distance of 6 feet was highly appreciated by study participants. The mean practice score was 5.22±5.11 (Range: 0–6) (Table 2).

Out of 377 women, only 91(24%) had excellent to good knowledge (Score: 9–12) and their attitude (score = 6), as well as practice score, was high (score: 5–6). Of 241(63.9%) had poor knowledge but their attitude and practice were above 90%. Two (0.5%) women had poor knowledge and negative attitude but the practice of these 2 women was appropriate (score = 4).

In Univariate analysis, regression coefficient showed that woman with age above 30 years (vs. ≤ 30, β = -2.87, p<0.01), grand multiparty (vs. no parity, β = -2.50, p<0.01), working women (vs. housewife, β = -4.29, p<0.01) and those women who had got information by

**Table 1. Characteristics of pregnant women [n = 377].**

| Variables | Frequency | Percentage |
|---|---|---|
| **Age Groups** | | |
| ≤ 30 | 208 | 55.2% |
| 31–40 | 161 | 42.7% |
| >40 | 8 | 2.1% |
| **Parity** | | |
| Zero | 88 | 23.3% |
| 2–4 | 144 | 38.2% |
| ≥5 | 145 | 38.5% |
| **Area of Resident** | | |
| Rural | 53 | 14.1% |
| Urban | 324 | 85.9% |
| Number of family members: | | |
| <4 | 155 | 41.1% |
| 4–7 | 176 | 46.7% |
| >7 | 46 | 12.2% |
| **Education status** | | |
| No formal education | 4 | 1.1% |
| Basic/Metric Level | 25 | 6.6% |
| College Level | 119 | 31.6% |
| University Level | 229 | 60.7% |
| **Occupational Status** | | |
| Housewife | 288 | 76.4% |
| Working women | 89 | 23.65 |
| **Sources of Information** | | |
| Health workers | 77 | 20.4% |
| Friends and relatives | 60 | 15.9% |
| Television | 98 | 26% |
| Internet | 142 | 37.75 |

internet (vs. others, β = -3.75, p<0.01) were significantly associated with low knowledge. In multivariate analysis, adjusted regression coefficient (by general linear model) showed that age above 30 years (vs. ≤ 30, β = -0.96, p<0.01), less family member<4 (vs. >7, β = -0.89, p<0.001), working women (vs. housewife, β = -1.58, p<0.01) and those women who had got information by internet (vs. others, β = -2.39, p<0.01) were significantly associated with lower knowledge score (Table 4).

Similarly, in univariate analysis for attitude and practice are showing that less than 30 years of age, nulliparous, less than 7 family members, basic level of education, housewives and sources of information by a health care worker, relatives, and television were significantly associated with a positive attitude and positive practice (Table 5).

## Discussion

To date, few studies have been conducted to assess knowledge, attitude, and practices among pregnant women against COVID 19 pandemic [15]. Despite the implementation of the COVID-19 vaccine, this pandemic not only poses a significant threat to public health but also created a lot of fear and anxiety among pregnant women. This directs the importance of awareness and adherence of pregnant women to preventive measures. Hence this might be the

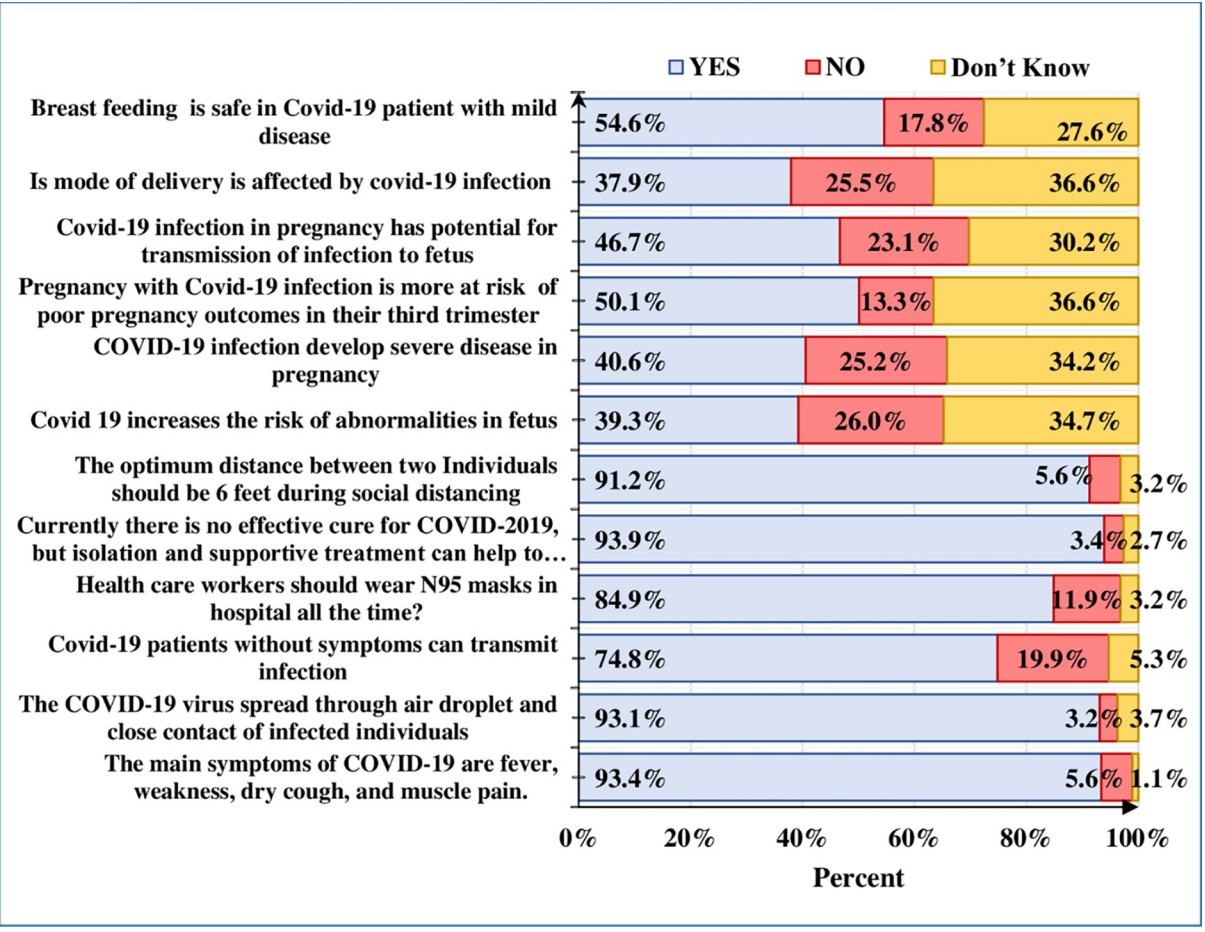

**Fig 1. Knowledge of pregnant women toward COVID-19. [n = 377].**

first study, conducted in our country to evaluate Knowledge, attitude, and practices among pregnant women against COVID-19 disease during the third wave of this pandemic.

Our study findings showed most pregnant women had poor knowledge regarding the impact of COVID-19 infection on pregnancy, although they were aware of symptomatology, the spread of infection, use of N-95 mask, social distancing and lack of optimal cure of the disease to date.

Despite their lack of knowledge, they were found to have a good attitude and appropriate practices against COVID-19 infection (Table 2).

In our study, less than 50% of respondents, were unaware of the impact of COVID-19 infection on pregnancy and its possible feto-maternal outcomes. About half of the women reported that they think breastfeeding is safe for mild diseases (Fig 1). Similar findings have been debated in the literature and raised a lot of concerns in pregnant women related to their

**Table 2. Mean KAP score of women toward COVID-19.**

| KAP | Mean ± SD | 95% CI for Mean | Min-Max |
|---|---|---|---|
| Knowledge Score | 6.63±2.48 | 6.38–2.48 | 0–11 |
| Attitude Score | 5.36±1.81 | 5.18–5.54 | 0–6 |
| Practice Score | 5.22±5.11 | 5.11–5.34 | 0–6 |

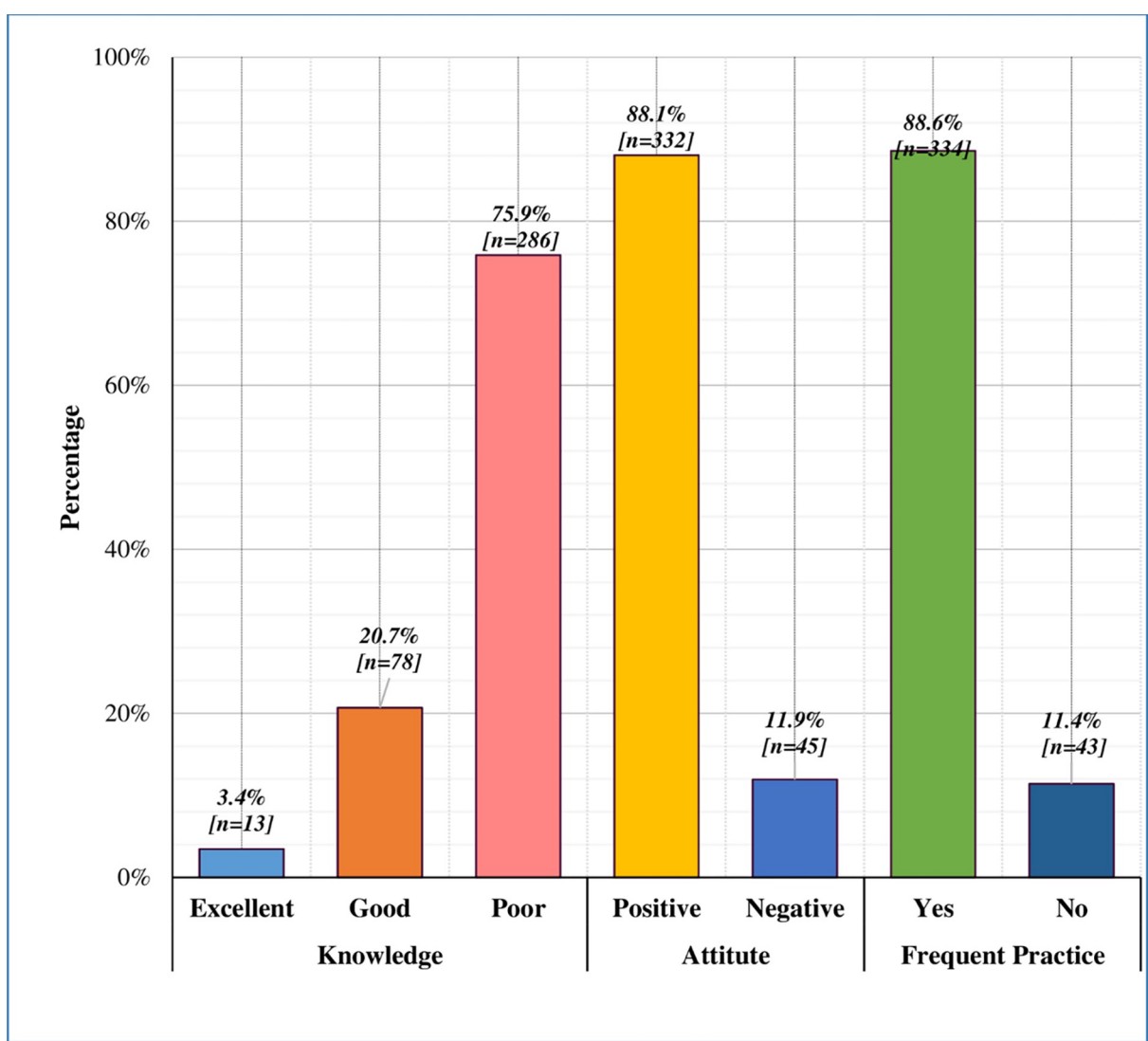

**Fig 2. Women's KAP toward COVID-19.**

pregnancy and unborn babies [16]. We propose that their concerns can be addressed by upgrading their knowledge regarding the impact of COVID-19 infection on pregnancy by taking comprehensive antenatal counseling sessions on booking antenatal visits and by sharing appropriate and reliable evidence of this pandemic correlating with pregnancy.

However, most of the obstetric participants (> 90%) correctly identified the symptoms, mode of transmission social distancing, and lack of specific treatment for this infection. (Fig 1). These findings are consistent with a study conducted in Bangladesh [17]. The probable reason for good awareness regarding these questions may be the mass media (Internet and television), as most of the participants (76.4%) disclosed the mass media as the commonest source of information and the second reason may be the demographic variation as most of the participants belongs to the urban region (85.9%) with a higher level of education (90%). These findings were comparable to the study performed in Kenya [18].

Univariate and multivariable linear regression analysis shows the factors; the age of more than 30 years, grand multiparity working women, and sources of information by internet

**Table 3. Women's attitude and practice related to COVID 19 [n = 377].**

| Attitude | Agree | Disagree | Undecided | No answer |
|---|---|---|---|---|
| Are you anxious about your safety? | 337 (89.4) | 31(8.2) | 5(1.3) | 4 |
| Are you anxious about the safety of your fetus? | 341 (90.5) | 17(4.5) | 15(4) | 4 |
| Do you have confidence that PPE (personal protection equipment) and contact precautions can protect you against COVID-19 infection? | 332 (88.1) | 36(9.5) | 0(0) | 9 |
| Practices | Yes | No | Sometimes | No answer |
| Do you use a face mask? | 374 (99.2) | 1(0.3) | 2(0.5) | 0 |
| Do you use gloves? | 180 (47.7) | 171 (45.4) | 25(6.6) | 1 |
| Do you use hand sanitizers? | 361 (95.8) | 7(1.9) | 9(2.4) | 0 |
| Do you use soap for hand washing? | 364 (96.6) | 7(1.9) | 3(0.8) | 3 |
| Are you avoiding large social gatherings | 356 (94.4) | 12(3.2) | 9(2.4) | 0 |
| Do you maintain a social distance of 6 feet with your peers? | 334 (88.6) | 17(4.5) | 26(6.9) | 0 |

**Table 4. Univariate and multivariable linear regression analysis showing the factors associated with the knowledge score of pregnant women related to COVID-19.**

| Variables | Unadjusted ß(SE) | P-value | Adjusted ß(SE) | P-Value |
|---|---|---|---|---|
| **Age Groups** | | | | |
| ≤ 30 | Ref | | Ref | |
| >30 | -2.87(0.21) | 0.0005 | -0.96(0.24) | 0.0005 |
| **Parity** | | | | |
| ≥5 | -2.50(0.23) | 0.0005 | -0.64(0.36) | 0.080 |
| 2–4 | 1.54(0.23) | 0.0005 | 0.31(0.16) | 0.049 |
| Zero | Ref | | Ref | |
| **Area of Resident** | | | - | - |
| Urban | -0.43(0.37) | 0.24 | | |
| Rural | Ref | | | |
| **Family members:** | | | | |
| <4 | 4.72(0.32) | 0.0005 | -0.89(0.27) | 0.001 |
| 4–7 | 4.87(0.31) | 0.0005 | 2.0(0.20) | 0.0005 |
| >7 | Ref | | Ref | |
| **Education status** | | | - | - |
| University/ College Level | -0.40(0.47) | 0.401 | | |
| No/Basic Level | Ref | | | |
| **Occupational Status** | | | | |
| Working women | -4.29(0.21) | 0.0005 | -1.58(0.19) | 0.0005 |
| Housewife | Ref | | Ref | |
| **Sources of Information** | | | | |
| Internet | -3.75(0.18) | 0.0005 | -2.39(0.34) | 0.0005 |
| Health Workers/TV/Relative | Ref | | Ref | |

ß (SE) = Regression coefficient (Standard error)

**Table 5. In Univariate analysis, factors associated with attitude and practice of pregnant women.**

| Variables | n | Attitude | | | Frequent Practice | | |
|---|---|---|---|---|---|---|---|
| | | Positive n = 332 | Negative n = 45 | P-Value | Yes n = 334 | No n = 43 | P-Value |
| **Age Groups** | | | | 0.0005 | | | 0.0005 |
| ≤ 30 | 208 | 208(100%) | 0 | | 208(100%) | 0 | |
| >30 | 169 | 124(73.4%) | 45(26.6%) | | 126(74.6%) | 43(25.4%) | |
| **Parity** | | | | 0.0005 | | | 0.0005 |
| Zero | 88 | 88(100%) | 0 | | 88(100%) | 0 | |
| 2–4 | 144 | 142(98.6%) | 2(1.4%) | | 144(100%) | 0 | |
| ≥5 | 145 | 102(70.3%) | 43(29.7%) | | 102(70.3%) | 43(29.7%) | |
| **Area of Resident** | | | | 0.001 | | | 0.001 |
| Rural | 53 | 53(100%) | 0 | | 53(10%) | 0 | |
| Urban | 324 | 279(86.1%) | 45(13.9%) | | 281(86.7%) | 43(13.3%) | |
| Number of family members: | | | | 0.0005 | | | 0.0005 |
| <4 | 155 | 155(100%) | 0 | | 155(100%) | 0 | |
| 4–7 | 176 | 176(100%) | 0 | | 176(100%) | 0 | |
| >7 | 46 | 1(2.2%) | 45(97.8%) | | 3(6.5%) | 43(93.5%) | |
| **Education status** | | | | 0.035 | | | 0.060 |
| University/ College Level | 348 | 303(87.1%) | 45(12.9%) | | 305(87.6%) | 43(12.4%) | |
| No/Basic Level | 29 | 29(100%) | 0(0%) | | 29(100%) | 0(0%) | |
| **Occupational Status** | | | | 0.0005 | | | 0.0005 |
| Housewife | 288 | 287(99.7%) | 1(0.3%) | | 287(99.7%) | 1(0.3%) | |
| Working women | 89 | 45(50.6%) | 44(19.7%) | | 47(52.8% | 42(47.2%) | |
| **Sources of Information** | | | | 0.0005 | | | 0.0005 |
| Internet | 142 | 97(68.3%) | 45(31.7%) | | 99(69.7%) | 43(30.3%) | |
| Health Workers/TV/Relative | 235 | 235(100%) | 0(0%) | | 235(100%) | 0(0%) | |

associated with poor knowledge score of pregnant women. (Table 4). These findings are consistent with one of the studies conducted in Nigeria [19].

Despite having poor knowledge regarding the impact of COVID 19 on pregnancy, study participants showed a positive attitude shown in Table 2 and Fig 2. This might be because of fear, anxiety, and concerns of pregnant women for their unborn babies, as most of them reported that they felt vulnerable and predominantly concerned about themselves (89.4%) and their unborn baby's health (90%). These findings are almost comparable to the study conducted in Iran and Malaysia [20, 21] Though these results contrast with another study that proposed negative emotions, anxiety, and panic affect their attitude [22].

In our study respondents showed good practices (88,6%) Fig 2, with a mean practice score of 5.22±5.11 (Range: 0–6) (Table 2). These findings are also comparable to other studies done in Pakistan that also revealed a good level of practice (88.7%) [23]. The reason of good practices might be the difference in educational status, as most of the respondents (92.3%) had tertiary level education. The only difference was the study participants, we recruited pregnant women, whereas health care workers were the participants in a comparable study. The other possible reasons for good preventive practices (mean of 88.6%) among pregnant women were the absence of high-quality evidence regarding the safety of vaccines during pregnancy, so the absence of high-quality evidence, that they are more inclined toward preventive practice.

Univariate analysis showed age, parity, type of residence, family size and mass media were found to be the significant predictors of positive attitudes and good practices among pregnant women. (Table 5).

## Conclusion

Pregnant patients demonstrated inadequate knowledge regarding the impact of COVID-19 on pregnancy. However, they showed a positive attitude and good practices in preventive measures. This highlights the need for community-based health education for pregnant women for COVID-19 to improve knowledge on a constant basis and further studies are required to see the impact of COVID-19 on pregnancy and feto-maternal outcomes.

## Strengths and limitations

The strength of this study is that it is the first study to evaluate the KAP and its association with socio-demographic variables against coronavirus infection among pregnant mothers in Pakistan. As this study has been conducted in a single tertiary care center and targeted only pregnant women. Hence, the result of this study limits the generalization of the study findings. Further, a multicenter study may be considered to understand the pregnant women's KAP so that the results can be generalizable for an action plan.

## Supporting information

**S1 Data. Data sheet.**
(XLS)

## Acknowledgments

Declarations our gratitude and appreciation go to the data collectors, pregnant women participants, physicians, and authorities of Aga Khan University Hospital.

## Author Contributions

**Conceptualization:** Sumaira Naz.

**Data curation:** Shamila Saleem.

**Formal analysis:** Amir Raza.

**Methodology:** Sumaira Naz, Syeda Dur e Shawar.

**Project administration:** Sumaira Naz, Shamila Saleem.

**Supervision:** Ayesha Malik.

**Writing – original draft:** Sumaira Naz, Syeda Dur e Shawar, Shamila Saleem.

**Writing – review & editing:** Sumaira Naz, Syeda Dur e Shawar, Shamila Saleem.

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
