## [Decision Letter · Decision Letter 0]

22 Dec 2021

PONE-D-21-19464Knowledge, attitudes, and practices (KAP) towards COVID-19 pandemic among pregnant women in a tertiary hospital Karachi, PakistanPLOS ONE

Dear Dr. malik,

Thank you for submitting your manuscript to PLOS ONE. After careful consideration, we feel that it has merit but does not fully meet PLOS ONE’s publication criteria as it currently stands. Therefore, we invite you to submit a revised version of the manuscript that addresses the points raised during the review process.

ACADEMIC EDITOR: Please look into the reviewer comments and address them in your revision==============================

We look forward to receiving your revised manuscript.

Kind regards,

Prasenjit Mitra, MD, CBiol, MRSB, MIScT, FLS, FACSc, FAACC

Academic Editor

PLOS ONE

Journal Requirements:

2. Please include additional information regarding the survey or questionnaire used in the study and ensure that you have provided sufficient details that others could replicate the analyses. For instance, if you developed a questionnaire as part of this study and it is not under a copyright more restrictive than CC-BY, please include a copy, in both the original language and English, as Supporting Information. If the original language is written in non-Latin characters, for example Amharic, Chinese, or Korean, please use a file format that ensures these characters are visible.

3. Please state whether you validated the questionnaire prior to testing on study participants. Please provide details regarding the validation group within the methods section.

6. Please include a separate caption for each figure in your manuscript.

7. Please ensure that you refer to Figure 2 in your text as, if accepted, production will need this reference to link the reader to the figure.

8. Please include your tables as part of your main manuscript and remove the individual files. Please note that supplementary tables (should remain/ be uploaded) as separate "supporting information" files".

9. We note you have included a table to which you do not refer in the text of your manuscript. Please ensure that you refer to Table 1 in your text; if accepted, production will need this reference to link the reader to the Table.

Reviewers' comments:

Reviewer's Responses to Questions

**Comments to the Author**

1. Is the manuscript technically sound, and do the data support the conclusions?

Reviewer #1: Partly

Reviewer #2: Yes

2. Has the statistical analysis been performed appropriately and rigorously? 

Reviewer #1: No

Reviewer #2: Yes

3. Have the authors made all data underlying the findings in their manuscript fully available?

Reviewer #1: No

Reviewer #2: Yes

4. Is the manuscript presented in an intelligible fashion and written in standard English?

Reviewer #1: No

Reviewer #2: Yes

5. Review Comments to the Author

Reviewer #1: 1. Sample size calculated by WHO calculator is 380 assuming a response rate of 50%, confidence interval (CI) 95%, Z as 1.96, and margin of error d as 5% by assuming 5000 deliveries occurred per annum in obstetrics and gynecology unit as per unit annual statistics – VERIFY AND INCLUDE THE CALCULATION IN THE TEXT

2. Explain the sampling method

3. Individual scoring of 10 and above were categorized as excellent whereas score below 10 were considered as poor knowledge – HOW WAS THE CUT OFF DETERMINED

4. QUESTIONNAIRE – 21 ITEM QUESTIONNAIRE – Explain the validation process

5. The attitude section consisted of three questions, and the response of each item collected on a 3 point Likert scale as follows 0 (“Disagree”), 1 (“Undecided”), and 2 (“Agree”) – UNDECIDED CATEROGRY IN LIKERT SCALE IS NOT UNIVERSALLY FOLLOWED, BETTER TO BE CONVERTED AS NEAUTRAL

6. The total score ranged from 0 to 6 with an overall greater score indicates more positive attitudes towards the COVID-19. A cut off level of ≥ 5 was set for more positive attitudes towards the prevention of COVID infection – HOW WAS CUT OFF ARRIVED?

7. The reliability of the questionnaire checked by conducting a pretest among pregnant women by taking 5% of the sample size. From the pretest, understandability, clarity, and organization of the questionnaire will be checked and reviewed – IF RELIABALITY WAS DONE – WHATS THE CRONBACH’S ALPHA VALUE AND ITEM ANALYSIS TO BE EXPLAINED

8. STUDY TOOL – The 21 item questionnaire – Validation process is not explained in the article.

9. Data was entered and analyzed using SPSS version 21. Data was analyzed by using SPSS 19. – DESCRIBE THE VERSION CLEARLY AND MENTION THE LICENSE AGREEMENT

10. Calculated sample size is 380, but the results were given only for 377 due to 3 missing data which is less than optimum sample size – Author has to view this seriously as sampling and sample size determines the internal and external validity of any quantitative study.

Reviewer #2: Summary:

The study is a well-grounded scientific research to assess the Knowledge, attitude and practice regarding COVID 19 among pregnant mothers. The findings of the study will be useful in identifying and filling the gaps in knowledge regarding COVID 19 among pregnant mothers.

Background:

1. WHO, ARDS and KAP has to be expanded for the first time in the manuscript before using them as abbreviations thereafter.

2. Since study has been completed the phrase “study has been planned” can be avoided. Instead “Study was done” can be used.

3. Grammatical errors are present throughout the background section which needs to be addressed.

4. COVID – 19 has to be represented uniformly in capital letters throughout the manuscript.

5. The Justification for the study could be explained in more detail with detailed statistics regarding COVID 19 related deaths among preganant mothers - Globally, Nationally and in the area of study.

Methods:

1. More explanation is required regarding the sample size calculation. (Eg: Reference for WHO sample size calculator)

2. Since the questionnaire is self-made and not standardised, Cronbach alpha and rationale behind choosing the cut-off has to be mentioned in the manuscript. The author could explain whether they used median/mean or interquartile range of the scores to choose the cut-offs.

3. In page 5, line 115, terms like “will be checked and reviewed” has to be avoided as the study is completed.

4. Sampling method and study duration are not mentioned in the methods section.

5. The author could provide details regarding the language in which questionnaire was used.

6. Details regarding the mode of interview could be provided by the author. They could mention whether the participants filled on their own or an investigator asked questions and filled the form.

Results:

1. In the results section, Table 1 could to be cited in Page 6, Line no:115 to give readers that the details are present in frequency table.

2. In figure 1, the numbers looks pixelated and not visible. The authors could provide an image with good clarity so that, the numbers looks readable.

3. The authors could give details regarding the participants who gave no answer in attitude and practice section and how those responses were used in analysis.

4. The authors could interpret the beta coefficients in details to add value to the interpretation of the findings.

5. In table 5, since 0 is present in many of the cells, the authors could give details on the statistical test in the univariate analysis used to establish association between variables.

Discussion:

1. Repetition of findings mentioned in the results could be avoided in the discussion. Instead a gist can be given and discussed in detail with relevant study comparisons.

6. PLOS authors have the option to publish the peer review history of their article (what does this mean?). If published, this will include your full peer review and any attached files.

Reviewer #1: No

Reviewer #2: No

---

## [Author Response · Author response to Decision Letter 0]

22 Jun 2022

RESPONSE TO REVIEWER

Manuscript style including file name is corrected

2. Please include additional information regarding the survey or questionnaire used in the study and ensure that you have provided sufficient details that others could replicate the analyses. For instance, if you developed a questionnaire as part of this study and it is not under a copyright more restrictive than CC-BY, please include a copy, in both the original language and English, as Supporting Information. If the original language is written in non-Latin characters, for example Amharic, Chinese, or Korean, please use a file format that ensures these characters are visible.

Data was collected on the self-administered questionnaire, developed after a literature review and from WHO recommendation [ ]. The content of the questionnaire was grouped into various themes, including demographic characteristics, obstetric variables, knowledge, attitude, and practice of COVID-19-related questions.

3. Please state whether you validated the questionnaire prior to testing on study participants. Please provide details regarding the validation group within the methods section.

A pretest was conducted to ascertain the validity of the questionnaire. Mentioned in manuscript

Link of Data set is created in Manuscript before references

5. Please include your full ethics statement in the ‘Methods section of your manuscript file. In your statement, please include the full name of the IRB or ethics committee who approved or waived your study, as well as whether or not you obtained informed written or verbal consent. If consent was waived for your study, please include this information in your statement as well.

Data was collected after approval of the ethical review committee (Reference number; 2020-4976-14204) of (Aga khan university hospital (AKUH) Karachi, Pakistan. Anonymity and confidentiality of data were maintained.

Data were collected during the full quarantine in March and April 2020. 

Pregnant patient attending obstetric clinics for consultation were asked to participation in the study, who agreed for participation an informed consent in writing has been taken before completing questionnaire. 

6. Please include a separate caption for each figure in your manuscript.

 Incorporated in manuscript

7. Please ensure that you refer to Figure 2 in your text as, if accepted, production will need this reference to link the reader to the figure.

Incorporated in manuscript

8. Please include your tables as part of your main manuscript and remove the individual files. Please note that supplementary tables (should remain/ be uploaded) as separate "supporting information" files".

9. We note you have included a table to which you do not refer in the text of your manuscript. Please ensure that you refer to Table 1 in your text; if accepted, production will need this reference to link the reader to the Table.

Incorporated in manuscript

Reviewers' comments:

Reviewer's Responses to Questions

Comments to the Author

1. Is the manuscript technically sound, and do the data support the conclusions?

Reviewer #1: Partly

Reviewer #2: Yes

2. Has the statistical analysis been performed appropriately and rigorously?

Reviewer #1: No

Reviewer #2: Yes

5. Have the authors made all data underlying the findings in their manuscript fully available?

Provided in Covering letter and as a supplementary file with name of S1 Data sheet

Reviewer #1: No

Reviewer #2: Yes

4. Is the manuscript presented in an intelligible fashion and written in standard English?

Reviewer #1: No

Reviewer #2: Yes

5. Review Comments to the Author

Reviewer #1: 1. Sample size calculated by WHO calculator is 380 assuming a response rate of 50%, confidence interval (CI) 95%, Z as 1.96, and margin of error d as 5% by assuming 5000 deliveries occurred per annum in obstetrics and gynecology unit as per unit annual statistics – VERIFY AND INCLUDE THE CALCULATION IN THE TEXT

1. The sample size calculated by the WHO calculator is 360 assuming a response rate of 50%, confidence interval (CI) of 95%, Z as 1.96, and margin of error d as 5% by assuming 5000 deliveries occurred per annum in obstetrics and gynecology unit as per unit annual statistics. Hence, the sample size was n= (Z) 2 P (1-P) N /d2 (N-1) + (Z) 2 P (1-P), by considering the incomplete responses, we included the target sample of 380. 2. 

2. Explain the sampling method

Non-Probability convenience sampling method

3. Individual scoring of 10 and above were categorized as excellent whereas score below 10 were considered as poor knowledge – HOW WAS THE CUT OFF DETERMINED

Above and equal to 80% (10/12) was considered excellent to good knowledge. 

4. QUESTIONNAIRE – 21 ITEM QUESTIONNAIRE – Explain the validation process

The validity of content was reviewed for appropriateness, relevance, applicability, and accuracy by two senior faculty members, epidemiologist and educationist. Long statements were rephrased and make them clear, and unambiguous. Items that need exclusion were highlighted and removed. 

5. The attitude section consisted of three questions, and the response of each item collected on a 3 point Likert scale as follows 0 (“Disagree”), 1 (“Undecided”), and 2 (“Agree”) – UNDECIDED CATEROGRY IN LIKERT SCALE IS NOT UNIVERSALLY FOLLOWED, BETTER TO BE CONVERTED AS NEAUTRAL

UNDECIDED have been replaced into Neutral 

6. The total score ranged from 0 to 6 with an overall greater score indicates more positive attitudes towards the COVID-19. A cut off level of 6 was set for more positive attitudes towards the prevention of COVID infection – HOW WAS CUT OFF ARRIVED?

Above and equal to 80% (5/6) was considered good attitude.

7. The reliability of the questionnaire checked by conducting a pretest among pregnant women by taking 5% of the sample size. From the pretest, understandability, clarity, and organization of the questionnaire will be checked and reviewed – IF RELIABALITY WAS DONE – WHATS THE CRONBACH’S ALPHA VALUE AND ITEM ANALYSIS TO BE EXPLAINED

Incorporated in methodology

Cronbach’s Alpha for Knowledge = 0.93 (12 item)

Cronbach’s Alpha for attitude = 0.98 (3 item)

Cronbach’s Alpha for practice = 0.85(6 item)

8. STUDY TOOL – The 21 item questionnaire – Validation process is not explained in the article.

Same as question 4

9. Data was entered and analyzed using SPSS version 21. Data was analyzed by using SPSS 19. – DESCRIBE THE VERSION CLEARLY AND MENTION THE LICENSE AGREEMENT

SPSS 19 version Used 

10. Calculated sample size is 380, but the results were given only for 377 due to 3 missing data which is less than optimum sample size – Author has to view this seriously as sampling and sample size determines the internal and external validity of any quantitative study.

Also Incorporated in manuscripts 

The sample size calculated by the WHO calculator is 360 assuming a response rate of 50%, confidence interval (CI) of 95%, Z as 1.96, and margin of error d as 5% by assuming 5000 deliveries occurred per annum in obstetrics and gynecology unit as per unit annual statistics. Hence, the sample size was n= (Z) 2 P (1-P) N /d2 (N-1) + (Z) 2 P (1-P), by considering the incomplete responses, we included the target sample of 380.

Reviewer#2: Summary:

The study is a well-grounded scientific research to assess the Knowledge, attitude and practice regarding COVID 19 among pregnant mothers. The findings of the study will be useful in identifying and filling the gaps in knowledge regarding COVID 19 among pregnant mothers.

Background:

1. WHO, ARDS and KAP has to be expanded for the first time in the manuscript before using them as abbreviations thereafter.

Corrected

2. Since study has been completed the phrase “study has been planned” can be avoided. Instead “Study was done” can be used.

Corrected

3. Grammatical errors are present throughout the background section which needs to be addressed.

Corrected

4. COVID – 19 has to be represented uniformly in capital letters throughout the manuscript.

Corrected

5. The Justification for the study could be explained in more detail with detailed statistics regarding COVID 19 related deaths among preganant mothers - Globally, Nationally and in the area of study.

Mentioned in manuscript 

Methods:

1. More explanation is required regarding the sample size calculation. (Eg: Reference for WHO sample size calculator)

Mentioned above and Incorporated in manuscript.

2. Since the questionnaire is self-made and not standardized, Cronbach alpha and rationale behind choosing the cut-off has to be mentioned in the manuscript. The author could explain whether they used median/mean or interquartile range of the scores to choose the cut-offs.

Incorporated in manuscript

3. In page 5, line 115, terms like “will be checked and reviewed” has to be avoided as the study is completed.

Done

4. Sampling method and study duration are not mentioned in the methods section.

Mentioned above and in manuscript

5. The author could provide details regarding the language in which questionnaire was used.

The questionnaire was prepared in English and then translated into the Urdu language with the assistance of language experts. Mentioned in manuscript

6. Details regarding the mode of interview could be provided by the author. They could mention whether the participants filled on their own or an investigator asked questions and filled the form.

Participant filled themselves. And one of the team members was available to facilitate the patients in case of queries related to the questionnaire. Mentioned in manuscript

Results:

1. In the results section, Table 1 could to be cited in Page 6, Line no:115 to give readers that the details are present in frequency table.

Corrected in manuscript

2. In figure 1, the numbers looks pixelated and not visible. The authors could provide an image with good clarity so that, the numbers looks readable.

Corrected in manuscript 

3. The authors could give details regarding the participants who gave no answer in attitude and practice section and how those responses were used in analysis.

Participant who did not respond were used in denominator in analysis and listed in table 3. For attitude, we observed 9 participants in which 4 did not respond in all three questions and five participants did not response 1 out of 3 questions of attitude. 

For Practice, we observed three participants in which two participants did not response 1 out of 5 questions and one participants did not response 2 out of 5 question. See table 3, we included in the analysis 

4. The authors could interpret the beta coefficients in details to add value to the interpretation of the findings.

Following lines also incorporated in results text

In Univariate analysis, regression coefficient showed that woman with above 30 years of age(vs. ≤ 30, β= -2.87, p<0.01), grand multiparty (vs. no parity, β= -2.50, p<0.01), working women (vs. housewife, β= -4.29, p<0.01) and those women who had got information by internet ( vs. others, β = -3.75, p<0.01) were significantly associated with low knowledge. In multivariate analysis, adjusted regression coefficient (by general linear model) showed that above 30 years (vs. ≤ 30, β= -0.96, p<0.01) , less family member<4 (vs. >7, β= -0.89, p<0.001), working women (vs. housewife, β= -1.58, p<0.01) and those women who had got information by internet ( vs. others, β = -2.39, p<0.01) were significantly associated with lower knowledge score (table 4). Corrected in manuscript also. 

5. In table 5, since 0 is present in many of the cells, the authors could give details on the statistical test in the univariate analysis used to establish association between variables

Corrected in manuscript: Chi-square test or fisher’s exact test was used for Univariate analysis. . P≤0.05 was considered as significant. 

Discussion:

1. Repetition of findings mentioned in the results could be avoided in the discussion. Instead a gist can be given and discussed in detail with relevant study comparisons.

Corrected in manuscript

6. PLOS authors have the option to publish the peer review history of their article (what does this mean?). If published, this will include your full peer review and any attached files.

Do you want your identity to be public for this peer review? For information about this choice, including consent withdrawal, please see our Privacy Policy.

Reviewer #1: No

Reviewer #2: No

Corrected in manuscript.

---

## [Decision Letter · Decision Letter 1]

25 Aug 2022

Knowledge, attitudes, and practices (KAP) towards COVID-19 pandemic among pregnant women in a tertiary hospital Karachi, Pakistan

PONE-D-21-19464R1

Dear Dr. malik,

We’re pleased to inform you that your manuscript has been judged scientifically suitable for publication and will be formally accepted for publication once it meets all outstanding technical requirements.

Kind regards,

Prasenjit Mitra, MD, CBiol, MRSB, MIScT, FLS, FACSc, FAACC

Academic Editor

PLOS ONE

Additional Editor Comments (optional):

Reviewers' comments:

Reviewer's Responses to Questions

**Comments to the Author**

1. If the authors have adequately addressed your comments raised in a previous round of review and you feel that this manuscript is now acceptable for publication, you may indicate that here to bypass the “Comments to the Author” section, enter your conflict of interest statement in the “Confidential to Editor” section, and submit your "Accept" recommendation.

Reviewer #1: All comments have been addressed

Reviewer #2: All comments have been addressed

2. Is the manuscript technically sound, and do the data support the conclusions?

Reviewer #1: Partly

Reviewer #2: Yes

3. Has the statistical analysis been performed appropriately and rigorously? 

Reviewer #1: Yes

Reviewer #2: Yes

4. Have the authors made all data underlying the findings in their manuscript fully available?

Reviewer #1: Yes

Reviewer #2: Yes

5. Is the manuscript presented in an intelligible fashion and written in standard English?

Reviewer #1: Yes

Reviewer #2: Yes

6. Review Comments to the Author

Reviewer #1: 1. Check and verify the references - whether its in recommended

2. The reference articles mentioned in the discussion part are not done in the pregnant women population for comparison except one study from africa

Reviewer #2: (No Response)

7. PLOS authors have the option to publish the peer review history of their article (what does this mean?). If published, this will include your full peer review and any attached files.

Reviewer #1: No

Reviewer #2: No

---

## [Editor Report · Acceptance letter]

16 Nov 2022

PONE-D-21-19464R1 

 Knowledge, attitudes, and practices (KAP) towards COVID-19 pandemic among pregnant women in a tertiary hospital in Karachi, Pakistan 

Dear Dr. Malik:

I'm pleased to inform you that your manuscript has been deemed suitable for publication in PLOS ONE. Congratulations! Your manuscript is now with our production department. 

Kind regards, 

on behalf of

Dr. Prasenjit Mitra 

Academic Editor

PLOS ONE